theoretical biology/biomathematics/health and disease and epidemiology

superinfection, asymptomatic infectious stage, pathogen evolutionary dynamics, life histories of pathogens

**Author for correspondence:**
Chadi M. Saad-Roy
e-mail: csaadroy@princeton.edu

# Superinfection and the evolution of an initial asymptomatic stage

Chadi M. Saad-Roy[1], Bryan T. Grenfell[2,3,5],
Simon A. Levin[2], Lorenzo Pellis[6,7], Helena B. Stage[6],
P. van den Driessche[8] and Ned S. Wingreen[1,4]

[1]Lewis-Sigler Institute for Integrative Genomics, [2]Department of Ecology and Evolutionary Biology, [3]Princeton School of Public and International Affairs, and [4]Department of Molecular Biology, Princeton University, Princeton, NJ, USA
[5]Division of International Epidemiology and Population Studies, Fogarty International Center, National Institutes of Health, Bethesda, MD, USA
[6]Department of Mathematics, University of Manchester, Manchester, UK
[7]The Alan Turing Institute, London, UK
[8]Department of Mathematics and Statistics, University of Victoria, Victoria, British Columbia, Canada

CMS-R, 0000-0002-2217-3071; LP, 0000-0002-3436-6487;
HBS, 0000-0001-9938-8452

Pathogens have evolved a variety of life-history strategies. An important strategy consists of successful transmission by an infected host before the appearance of symptoms, that is, while the host is still partially or fully asymptomatic. During this initial stage of infection, it is possible for another pathogen to superinfect an already infected host and replace the previously infecting pathogen. Here, we study the effect of superinfection during the first stage of an infection on the evolutionary dynamics of the degree to which the host is asymptomatic (host latency) in that same stage. We find that superinfection can lead to major differences in evolutionary behaviour. Most strikingly, the duration of immunity following infection can significantly influence pathogen evolutionary dynamics, whereas without superinfection the outcomes are independent of host immunity. For example, changes in host immunity can drive evolutionary transitions from a fully symptomatic to a fully asymptomatic first infection stage. Additionally, if superinfection relative to susceptible infection is strong enough, evolution can lead to a unique strategy of latency that corresponds to a local fitness minimum, and is therefore invasible by nearby mutants. Thus, this strategy is a branching point, and can lead to coexistence of pathogens with different latencies. Furthermore, in this new framework with superinfection, we

also find that there can exist two interior singular strategies. Overall, new evolutionary outcomes can cascade from superinfection.

## 1. Introduction

Annually, infectious diseases lead to substantial morbidity and mortality in human populations. To limit this burden and successfully decrease disease transmission, it is important to quantify epidemiological dynamics (e.g. [1–3]). Numerous factors influence transmission and are thus important to consider in epidemiological models. For example, the ability of a pathogen to transmit while a host remains without symptoms has often been highlighted in the epidemiological literature as an important characteristic (e.g. [4–6]). In particular, if hosts are able to transmit an infection while initially remaining asymptomatic, or 'fully latent', and thereby escaping detection, control measures become substantially more difficult [5]. Therefore, understanding the dynamics of a sub-symptomatic or fully asymptomatic first infection stage is imperative for disease control. The importance of transmission without symptoms is strikingly illustrated by the recent COVID-19 pandemic [7–11].

The evolutionary dynamics of a pathogen play a central role in host epidemiology. Evolving pathogens are under a series of constraints, including those imposed by within-host replication and host immunity. These constraints can modulate the degree of latency, i.e. the spectrum from fully symptomatic to fully asymptomatic, in the initial infectious stage. Here, we consider the initial infectious stage to be one where transmission may occur; this stage is distinct from and may follow a stage after infection where hosts are infected but not infectious. In other words, we consider a two-stage infection process where the first stage is said to have 'zero latency' if this stage is fully symptomatic, whereas we say the first stage has 'maximal latency' if it is fully asymptomatic. Thus, latency is a measure of the degree to which a host in the initial infectious stage fails to exhibit symptoms. If a host is more latent in the first stage (i.e. more asymptomatic), then the pathogen can potentially remain in this stage longer due to lowered host immune responses that would lead to pathogen clearance. However, in order for the pathogen to successfully 'hide' from the host immune system, decreased within-host replication and thus decreased pathogen shedding is required, which in turn decreases transmission (see [12] for a detailed discussion and further references). Therefore, there can exist a trade-off between transmission and progression in the first stage of an infection.

Pathogen evolutionary dynamics can be examined through a variety of approaches. Historically, there has been strong interest in understanding pathogen virulence from an evolutionary perspective (e.g. [2,13–20]). With regard to latency, Saad-Roy et al. [12] coupled a simple epidemiological model with two infection stages to a life-history approach to study the evolutionary outcomes of latency ($\lambda$) on the initial infection stage. These authors examined simple forms of progression and transmission trade-offs, and found four possible evolutionary behaviours: a unique evolutionarily stable strategy (ESS) at zero latency, positive latency, maximal latency or bistability with zero and maximal latency. With more complicated trade-off forms that changed concavity of progression and transmission as functions of latency, these authors also observed the appearance of multiple interior evolutionarily singular strategies. Throughout their analyses, Saad-Roy et al. [12] assumed that a pathogen could not infect a host currently infected with, or with immunity to, another strain of the pathogen.

The infection of an already-infected host by a different strain has been suggested as an important mechanism in infectious disease dynamics [21,22], in particular in the evolution of human immunodeficiency virus (HIV) and certain variants of toxoplasma [23]. Empirically, successful infections of already-infected hosts by other strains have been observed, for example for HIV [24], and for hepatitis C virus (HCV) [25]. Thus, incorporating such processes in evolutionary-epidemiological models is important.

Here, we follow [21,26–29] and define *superinfection* as the within-host competitive clearance of the resident infecting pathogen by another pathogen. Therefore, superinfection is distinct from co-infection, where a host can be simultaneously infected by multiple strains [21,30].

The ties between superinfection and the evolution of virulence have been explored through various modelling frameworks (e.g. [13,27–29,31,32]). Notably, Levin & Pimentel [13] formulated a model with virulent and avirulent strategies, where superinfection only occurs so that a virulent pathogen can clear an avirulent pathogen, and these authors proved that coexistence with both strategies is possible (see also Levin [33]). Nowak & May [29] showed the possibility of coexistence of pathogens with different virulences. Furthermore, as a function of virulence difference between invader and resident, Gandon et al. [27] revealed that a fitness minimum (for the pathogen) can exist if the increase in superinfection propensity is rapid enough. In addition, Gandon et al. [28] examined the coevolution of pathogen virulence and host resistance in a framework that includes superinfection. More generally, Alizon & van

Baalen [34] used a susceptible–infectious (SI) epidemiological model with multiple strains and within-host dynamics to study the effect of multiple infections on the evolution of virulence.

With a transmission–virulence trade-off, Boldin & Diekmann [35] and Boldin *et al*. [36] have thoroughly examined the role of superinfection for evolutionary attraction towards a fitness minimum (i.e. branching) in models without recovery and thus with lifelong infection. Boldin & Diekmann [35] tied their evolutionary analyses to within-host dynamics, and found that branching can occur for specific dependencies of superinfection on resident and mutant pathogens, leading to epidemiological and evolutionary coexistence of different strains.

Other studies have examined superinfection and pathogen evolution more generally. Without a transmission–virulence trade-off, Alizon [37] showed that branching can occur as long as there is a strong asymmetry in the propensity for invading (and thus replacing) versus being invaded (and thus replaced) for a pathogen as a function of virulence. Kada & Lion [38] incorporated superinfection in a virulence–recovery coevolutionary model [39], and found that strong enough superinfection leads to virulent pathogens and immune-capable hosts.

In this paper, we examine the effect of superinfection on the evolutionary dynamics of latency in the first infectious stage. Since minimizing presymptomatic transmission is necessary for adequate control [5], understanding how and why superinfection is tied to the evolution of latency in this stage could inform public health. In particular, characterizing the possible evolutionary outcomes of latency for different regimes of superinfection could highlight the degree of latency that could arise from the various evolutionary scenarios. In turn, parameter regions where a fully asymptomatic first stage of infection is evolutionarily stable could be identified. Lastly, determining the importance of host immunity duration following recovery and the interplay with superinfection would emphasize the relative effects that host immunity can have on the evolutionary dynamics of latency. We focus on obligate single host, directly transmitted pathogens, such as measles, smallpox, norovirus, HIV, herpes simplex virus, etc. (See [12] for a detailed discussion of the ties between our underlying model, latency and viruses that infect humans.)

Motivated by the underlying immune trade-off in the first infection stage, we assume that this first stage is susceptible to superinfection, after which pathogen loads become too high and host immune responses too strong for another pathogen to successfully infect and clear the existing infection. Under these assumptions, we study life-history strategies of pathogens in terms of latency during an initial infection stage, as defined in [12]. We show that the inclusion of superinfection qualitatively and quantitatively affects the evolutionary dynamics of latency. In particular, we highlight how the interplay of host immunity and superinfection can modulate pathogen evolutionary outcomes. We also find that superinfection mediates the existence of a unique branching point, i.e. an evolutionary attractor that does not maximize fitness but instead is a local fitness minimum and is therefore evolutionarily unstable. Thus, this branching point can lead to the mutual invasibility of strains with different latencies, in particular what numerically appears to be an evolutionarily stable coalition of strains at zero and maximal (fully asymptomatic) latency. However, when there is mutual invasibility, it remains an open problem to show whether the result is always stable coexistence. We also illustrate other novel evolutionary behaviours that emerge due to superinfection.

## 2. Model preliminaries

We extend the analysis of [12] by assuming that in the initial stage of infection, any infected host is subject to superinfection by a different strain with some likelihood relative to susceptible infection. Throughout our analyses, we assume that the superinfecting strain immediately replaces the previous strain within a single host. Thus, while the evolutionary analysis in [12] relied upon the minimization of the susceptible fraction, i.e. the 'resource' (see Tilman [40] for more details), considering superinfection requires a different approach. Here, we use adaptive dynamics [26,41–44] to identify evolutionarily singular strategies and long-term evolutionary dynamics.

The basic SIIRS model, which will be a starting point for our extensions, is [12]

$$
\begin{aligned}
\frac{dS}{dt} &= \delta - \alpha_1 SI_1 - \alpha_2 SI_2 - \delta S + \mu R, \\
\frac{dI_1}{dt} &= \alpha_1 SI_1 + \alpha_2 SI_2 - (\nu_1 + \delta)I_1, \\
\frac{dI_2}{dt} &= \nu_1 I_1 - (\nu_2 + \delta)I_2
\end{aligned}
$$

and

$$
\frac{dR}{dt} = \nu_2 I_2 - (\mu + \delta)R.
$$

(2.1)

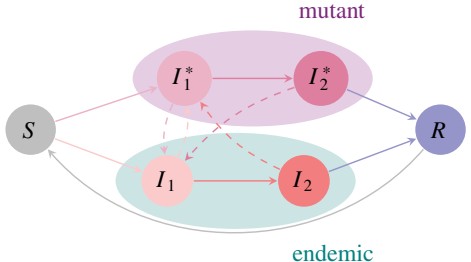

**Figure 1.** Formulation of the SIIRS model that includes superinfection, with the flowchart adapted from [12] to include superinfection dynamics. Note that the dashed arrows do not represent progression, but rather denote superinfection processes. Invasion dynamics are depicted in this flowchart through the mutant and resident types competing both for susceptibles and individuals that are in the first infection stage.

In this model, the population size is constant and $S$, $I_1$, $I_2$ and $R$ denote the fractions of individuals that are susceptible, in the first infection stage, in the second infection stage and recovered, respectively. Furthermore, $\delta$ denotes the death rate as well as the compensating birth rate, $\alpha_i$ is the transmission rate, $v_i$ is the progression rate of stage $i$, for $i = 1, 2$, and $\mu$ is the rate of loss of immunity. For the SIIRS epidemiological model, there is a unique endemic equilibrium if the basic reproduction number [45–47] is greater than one, and this unique equilibrium is locally stable [12]. Thus, these results enable us to use the adaptive dynamics approach on this system. Otherwise, if the basic reproduction number is less than one, the disease-free equilibrium is globally asymptotically stable [12]; in other words, the disease dies out.

Figure 1 summarizes the model that includes superinfection. To include superinfection in the model framework, we denote by $I_1^*$ and $I_2^*$ the fraction of individuals infected with the mutant phenotype in the first and second infection stages, respectively, with the corresponding first stage transmission and progression rates denoted $\alpha_1^* = \alpha_1[\lambda^*]$ and $v_1^* = v_1[\lambda^*]$. We denote the relative likelihood of superinfection compared to susceptible infection, i.e. the 'superinfection parameter', of mutant $\lambda^*$ to resident $\lambda$ as $\sigma[\lambda, \lambda^*]$. As in Gandon *et al.* [26,28], we assume that the superinfection propensity is constant across phenotypic space, i.e. $\sigma[\lambda, \lambda^*] = \sigma$. If $\sigma < 1$, then superinfection is less likely than susceptible infection, whereas if $\sigma > 1$, then superinfection is more likely. Either of these cases may be more realistic than the other for specific situations. We formulate the rates of change over time of the fraction infected with the mutant phenotype in either infection stage as

$$\left.\begin{aligned}
\frac{\mathrm{d}I_1^*}{\mathrm{d}t} &= \alpha_1^* I_1^* S + \alpha_2 I_2^* S + \sigma\alpha_1^* I_1 I_1^* + \sigma\alpha_2 I_1 I_2^* - (v_1^* + \delta)I_1^* - \sigma(\alpha_1 I_1 I_1^* + \alpha_2 I_2 I_1^*) \\
\text{and} \qquad \frac{\mathrm{d}I_2^*}{\mathrm{d}t} &= v_1^* I_1^* - (v_2 + \delta)I_2^*.
\end{aligned}\right\} \tag{2.2}$$

Thus, we assume that immunity against the resident strain following recovery also bestows complete protection against the mutant. With the $I_1^*$ and $I_2^*$ equations, we use the next-generation matrix [46,47] at the equilibrium with no mutant, $I_1^* = I_2^* = 0$, to compute the basic reproduction number of a mutant pathogen with strategy $\lambda^*$ invading a population of the endemic strain $\lambda$ at the epidemiological equilibrium. This reproduction number is

$$\mathcal{R}_0[\lambda, \lambda^*]$$
$$= \left( \frac{\alpha_1[\lambda^*]}{v_1[\lambda^*] + \delta + \sigma(\alpha_1[\lambda]\widehat{I}_1[\lambda] + \alpha_2\widehat{I}_2[\lambda])} + \frac{v_1[\lambda^*]}{v_1[\lambda^*] + \delta + \sigma(\alpha_1[\lambda]\widehat{I}_1[\lambda] + \alpha_2\widehat{I}_2[\lambda])} \frac{\alpha_2}{v_2 + \delta} \right) (\widehat{S}[\lambda] + \sigma\widehat{I}_1[\lambda]), \tag{2.3}$$

where $\alpha_1$, $v_1$ are functions of $\lambda$, $\widehat{S} = \dfrac{1}{\dfrac{\alpha_1}{v_1 + \delta} + \dfrac{v_1}{v_1 + \delta}\dfrac{\alpha_2}{v_2 + \delta}}$, $\widehat{I}_1 = \dfrac{\left(1 - \dfrac{1}{\dfrac{\alpha_1}{v_1 + \delta} + \dfrac{v_1}{v_1 + \delta}\dfrac{\alpha_2}{v_2 + \delta}}\right)}{1 + \dfrac{v_1}{v_2 + \delta} + \dfrac{v_2}{\delta + \mu}\dfrac{v_1}{v_2 + \delta}}$, and

$\widehat{I}_2 = \dfrac{v_1}{v_2 + \delta}\widehat{I}_1$. Here, $\mathcal{R}_0[\lambda, \lambda^*]$ can be interpreted biologically. The first and second terms are the numbers of new infections with the pathogen of phenotype $\lambda^*$ resulting from the initial mutant

infection in the first and second stages, respectively, $\sigma\alpha_1[\lambda]\widehat{I}_1[\lambda]$ and $\sigma\alpha_2[\lambda]\widehat{I}_2[\lambda]$ are the rates of superinfection of a mutant pathogen by the endemic strain in the first and second stages, respectively.

# 3. Results

## 3.1. General evolutionary dynamics

With no superinfection, Saad-Roy *et al.* [12] found that zero, partial or infinite latency are all possible unique evolutionary outcomes, in addition to a fourth case with bistability of zero and infinite latency types. Here, we follow their approach and assume that transmission $\alpha_1[\lambda]$ and progression $v_1[\lambda]$ both decay as functions of latency $\lambda$, so that

$$\alpha_1[\lambda] = b_1(F[\lambda])^{-b_2} + \alpha_{1,\infty} \tag{3.1}$$

and

$$v_1[\lambda] = c_1(F[\lambda])^{-c_2} + v_{1,\infty} \tag{3.2}$$

with $F[\lambda]$ such that $F[0] = 1$, $F[\infty] = \infty$ and $F'[\lambda] > 0$ for all $\lambda \geq 0$, and $b_1, b_2, c_1, c_2, \alpha_{1,\infty}, v_{1,\infty} \geq 0$. For clarity, we present the analyses for an exponential formulation, i.e. $F[\lambda] = e^{\lambda}$, in the electronic supplementary material (e.g. theorem 2), but equivalent results hold for the more general forms (electronic supplementary material, remark 5). Indeed, Saad-Roy *et al.* [12] highlighted that their results do not depend upon the exact parametrization of $\lambda$ but rather on the relationship between $\alpha_1$ and $v_1$ in addition to their values at maximal latency. Furthermore, to simplify our analyses, we take the biologically reasonable assumption that the basic reproduction number $k$ of the second infection stage is greater than 1, i.e. $k = \alpha_2/(v_2 + \delta) > 1$, where $k$ is the expected number of new infections that arise in a naive population from a single host in the second stage.

The ability of a pathogen to superinfect gives rise to some notable evolutionary differences with respect to the previous model with no superinfection [12]. The evolutionary dynamics can be partitioned based upon the value of the progression decay exponent, $c_2$, and the transmission decay exponent, $b_2$.

We first assume that transmission in the first stage decays faster than progression, i.e. $b_2 > c_2$. Thus, solving for $\alpha_1$ in terms of $v_1$ [12], $b_2 > c_2$ means that the dependence of $\alpha_1$ on $v_1$ is concave up. If there exists an interior evolutionarily singular strategy, then it is a fitness minimum and this is therefore not an ESS (electronic supplementary material, theorem 1).

The exact evolutionary dynamics depend upon the value of the superinfection parameter, $\sigma$, on the sign of $\partial\mathcal{R}_0/\partial\lambda^*$ at $\lambda = \lambda^* = 0$, and on the value of the fully latent transmission rate times the average host lifespan, $\alpha_{1,\infty}/\delta$. Here, we succinctly summarize the qualitative behaviours and focus on the transitions in evolutionary dynamics mediated by the superinfection parameter in our model. We describe the various conditions and thresholds for each evolutionary outcome in electronic supplementary material, *Expanded Results*.

For different regimes of the superinfection parameter, $\sigma$, our results are illustratively summarized in figure 2. Since we have assumed that superinfection occurs ($\sigma > 0$) only in the first infection stage, we let $\tau$ denote the average time a host infected by or with immunity to one pathogen cannot get superinfected, i.e. the average time that a host is not in $I_1$ after infection before return to susceptibility. Since $1/(v_2 + \delta)$ is the average time in $I_2$, $v_2/(v_2 + \delta)$ is the probability of surviving $I_2$, and $1/(\mu + \delta)$ is the average time in $R$, it follows that $\tau = 1/(v_2 + \delta) + (v_2/(v_2 + \delta))(1/(\mu + \delta))$. Thus, $\tau\delta$ is biologically intuitive and corresponds to the fraction of individuals that die while in $I_2$ or $R$. If the superinfection parameter is such that $\sigma < (1 - \tau\delta)/k$ (which forces $\sigma < 1$), then the possible evolutionary outcomes mirror almost exactly those without superinfection [12]. Indeed, there can either be a unique ESS at zero latency, locally stable ESSs at zero and maximal latency, or a locally stable ESS at maximal latency (figure 2a). Furthermore, for certain values of $\alpha_{1,\infty}/\delta$, there is either a unique or no interior evolutionarily singular strategy, whereas for other values the existence of additional singular strategies is possible.

If the likelihood of superinfection increases, so that $(1 + v_{1,\infty}\tau)/k > \sigma > (1 - \tau\delta)/k$, then a new evolutionary phenomenon emerges. For these values of superinfection, there can exist a *unique branching point* for intermediate values of the fully latent transmission rate times the average host lifespan, $\alpha_{1,\infty}/\delta$. This branching point is an evolutionarily singular strategy that is convergence stable [41], i.e. for residents with latency on either side of this singular strategy, mutants with latency intermediate to the singular strategy can invade, but the singular strategy is not evolutionarily stable.

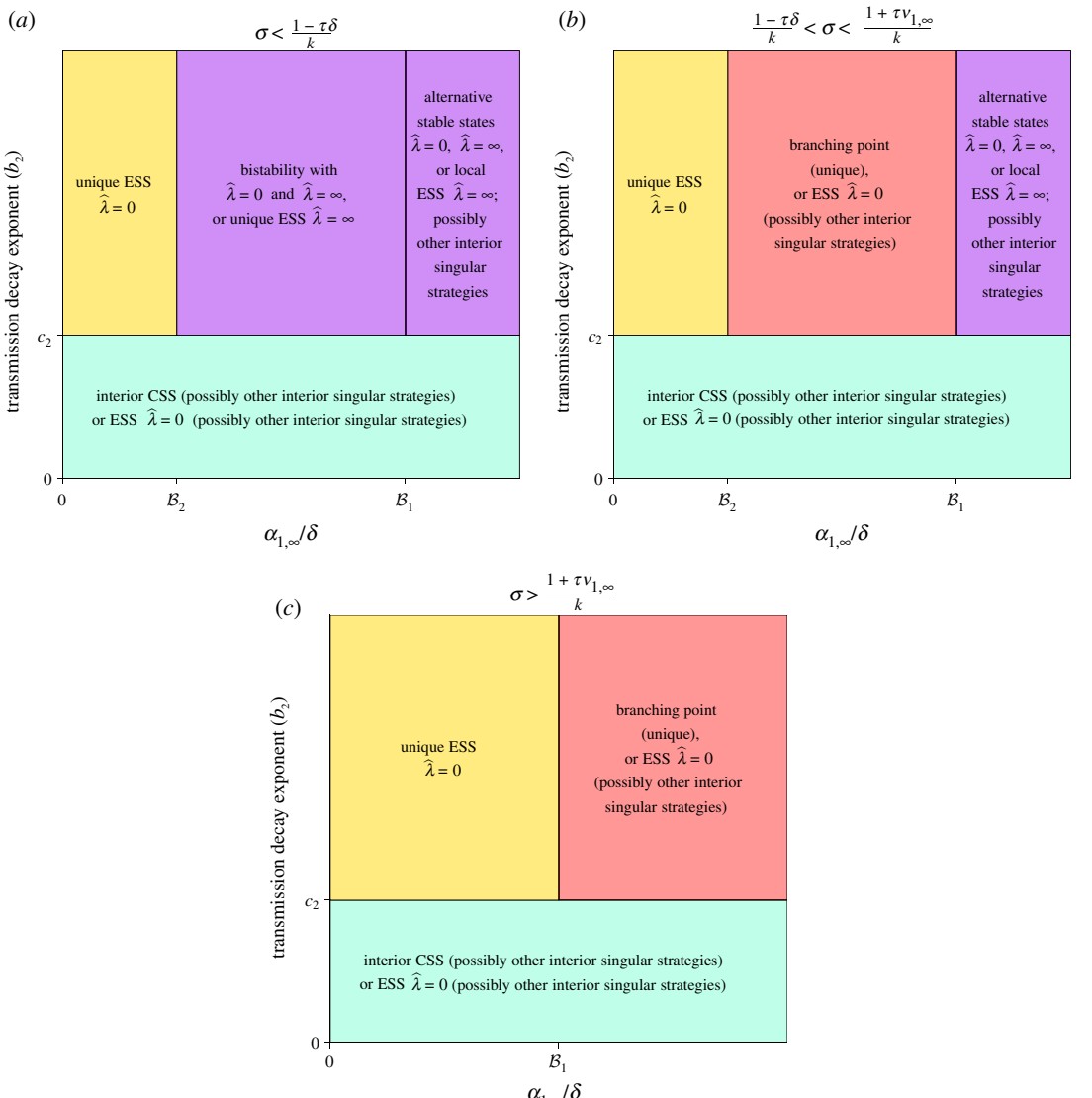

**Figure 2.** Schematics illustrating the possible evolutionary dynamics with superinfection. The evolutionary outcomes depend on the values of the superinfection parameter ($\sigma$), the transmission decay exponent ($b_2$), and the fully latent transmission rate ($\alpha_{1,\infty}$). Here, $\mathcal{B}_1 = k + (\sigma k(k-1))/\tau\delta$ and $\mathcal{B}_2 = k + [(\sigma k(k-1)(\nu_{1\infty} + \delta))/\delta]/(\tau\nu_{1,\infty} + 1 - k\sigma)$ (see electronic supplementary material, *Expanded Results*). (a) The superinfection parameter is small enough so that $0 < \mathcal{B}_2 < \mathcal{B}_1$. (b) The superinfection parameter is of intermediate value, giving $0 < \mathcal{B}_1 < \mathcal{B}_2$. (c) The superinfection parameter is larger than $(1 + \tau\nu_{1,\infty})/k$, which results in $\mathcal{B}_2 < 0 < \mathcal{B}_1$ (electronic supplementary material, remarks 3, 4). Here, ESS, evolutionarily stable strategy; CSS, continuously stable strategy.

Thus, if a strain with this strategy is endemic, mutants have higher fitness and can invade. This can lead to mutual invasibility of pathogens with different latencies (figure 2b and electronic supplementary material, *Expanded Results*). If the superinfection parameter increases further, so that $\sigma > (1 + \nu_{1,\infty}\tau)/k$, the possibility of alternative stable states at zero and maximal latency disappears: instead, there is either a branching point, or an ESS at zero latency (figure 2c, electronic supplementary material, remarks 3, 4).

If progression decays faster than transmission, i.e. $c_2 > b_2$, then $\alpha_1$ as a function of $\nu_1$ is concave down. Furthermore, any interior evolutionarily singular strategy is a local maximum of fitness and is thus an ESS (electronic supplementary material, theorem 1). In this setting, there exists either at least one interior attractor which is an interior ESS and is thus a continuously stable strategy (CSS), or there exists at least one local attractor at zero latency (electronic supplementary material, theorems 2, 4). The CSS is shown to be unique in certain special cases (electronic supplementary material, theorems 2, 4). These results qualitatively mirror those without superinfection, where $c_2 > b_2$ guarantees either a unique interior ESS or a global ESS at zero latency.

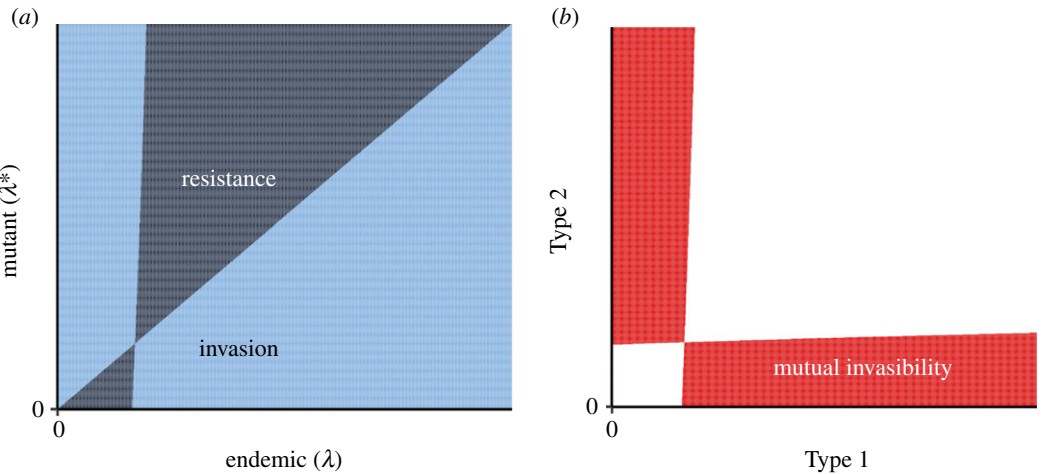

**Figure 3.** (a) Schematics for the existence of a unique branching point, i.e. an evolutionarily singular strategy that is convergence stable but not an ESS. Here, $\sigma = 1 < (1 + v_{1,\infty}\tau)/k$ so that the regime is as depicted in figure 2b, with $\mathcal{B}_1 < \alpha_{1,\infty}/\delta < \mathcal{B}_2$. Dark blue denotes the endemic strategy resisting mutant invasion, whereas light blue denotes mutant invading the endemic strategy. (b) Regions of mutual invasibility (depicted in red).

When transmission decays faster than progression, i.e. $b_2 > c_2$ (and $\alpha_1$ as a function of $v_1$ is concave up), it is also useful to consider the evolutionary outcomes for a fixed value of $\alpha_{1,\infty}/\delta$ as the superinfection parameter $\sigma$ varies. First, consider $\alpha_{1,\infty}/\delta$ close to zero (i.e. where there is a unique ESS at zero latency in figure 2a). Then, irrespective of the values of $\sigma$, there is a unique ESS at zero latency (figure 2, since $\mathcal{B}_1$ is an increasing function of $\sigma$). Second, consider a value of $\alpha_{1,\infty}/\delta$ where there is a local ESS at maximal latency (figure 2a) for small $\sigma$. As $\sigma$ increases, this local ESS at maximal latency disappears, and there is either an ESS at zero latency (which is either local or unique) or a unique branching point (figure 2b,c). The existence of a branching point significantly complicates the evolutionary dynamics of latency.

## 3.2. Implications of branching

In the previous section, we have found regions of parameter space where a convergence-stable singular strategy is not an ESS, which gives rise to the very interesting behaviour of branching. Furthermore, for those parameter regimes where we have proved that a branching point can exist, it is also unique. In figure 3a, we illustrate the existence of a unique evolutionarily singular strategy that is a branching point. From the pairwise invasibility plot (PIP), it is clear that this singular strategy is an attractor, but allows nearby mutants to invade. Thus, if initial latency is above or below the branching point, mutations intermediate to the initial strategy lead to invasion. As evolution progresses, the resident pathogen continues to approach the singular strategy. However, as the singular strategy is reached, mutants with smaller or larger latencies can invade, giving rise to branching and mutual invasibility. As an example, we schematically illustrate the region of mutual invasibility in figure 3b for two competing types when there exists a branching point.

Since our model assumes immediate replacement of the infecting strain by the superinfectant, the existence of a branching point in our model could be considered counterintuitive. This, despite the fact that a superinfectant can immediately take over a single host, does not prevent mutual invasibility of strains at the population level. In certain ecological models, mutual invasibility implies coexistence (see Chesson [48] and Geritz [49] for more details). It is possible, though we have not proved, that this result also holds for our model, and it numerically appears that coexistence with different strategies of latencies can occur for certain parameter values.

After branching, it is complicated to determine the evolutionary dynamics of latency. However, if an endemic equilibrium with two coexisting types exist, then a generalized adaptive dynamics approach applies in this case also (see [42]), and we have employed this framework to explore the evolutionary implications in our model (electronic supplementary material, *Evolutionary dynamics in the presence of coexistence*). In particular, we have shown that, if there exists any isocline for either coexisting type, then it is a fitness minimum. Thus, any interior evolutionarily singular coalition, if it exists, would be

evolutionarily unstable (electronic supplementary material, theorem 1). Furthermore, for the example in figure 3, we numerically find that a coalition with types at zero and infinite latency is evolutionarily stable.

The mutual invasibility of two different types of latencies has interesting analogies with the competitive-exclusion principle [40,50], and similar observations have been made for coexistence with superinfection in a model with one infection stage [37]. Indeed, with no superinfection, the only 'resource' for the pathogen is the susceptible population, and mutual invasibility between types is not possible. However, with superinfection, our model can be interpreted as having two 'resources' for pathogens: the susceptible population, and the population that is in the first infection stage.

## 3.3. The role of host immunity

In the model with no superinfection [12], the duration of immunity, $1/\mu$, following infection has no effect on the evolutionary dynamics of latency. Here, even with the simplest model of superinfection where the superinfecting pathogen immediately clears the 'resident' pathogen, the duration of immunity can have a dramatic effect on the possible evolutionary outcomes of latency.

To understand intuitively why the rate of waning immunity $\mu$ can affect the dynamics in our model, consider both extremal values, i.e. lifelong immunity ($\mu = 0$) and no immunity ($\mu = \infty$). If $\mu = 0$, then $\tau = 1/\delta$ is the average lifespan, so that $(1 - \tau\delta)/k = 0$. Thus, for any superinfection $\sigma > 0$, the resulting evolutionary dynamics are as in figure 2*b* or *c*. On the other hand, if $\mu = \infty$, then $\tau = 1/(v_2 + \delta)$ is the death-adjusted duration of the second stage, so that $(1 - \tau\delta)/k = (1/k)(v_2/(v_2 + \delta))$. Thus, for $0 < \sigma < (1/k)(v_2/(v_2 + \delta))$, the evolutionary dynamics will be different if $\mu = 0$ or if $\mu = \infty$. If there is no immunity, then the evolutionary dynamics are almost as in the case with no superinfection, whereas if there is lifelong immunity following infection, then the qualitative evolutionary dynamics have important differences. Notably, a branching point can emerge (figure 2*b*), or the possibility of alternative stable states can disappear (figure 2*c*).

Next, we numerically examine some possible evolutionary transitions due to host immunity. Figure 4 numerically illustrates the role that immunity can play in the evolutionary dynamics of latency. The most substantial differences in outcomes occur when the transmission decay exponent is greater than the progression decay exponent, i.e. $b_2 > c_2$, shown in figure 4*a,b*. With lifelong immunity, i.e $\mu = 0$, for this example it appears that there is a unique ESS is at $\hat{\lambda} = \infty$. As the duration of immunity is decreased, an unstable evolutionarily singular strategy emerges. If this singular strategy is convergence stable, then it is a branching point, and there is mutual invasibility. If this singular strategy is not convergence stable, then it gives rise to bistability with zero and infinite latency. As the duration of immunity is further decreased to zero, i.e. $\mu \to \infty$, then there is a unique ESS at $\hat{\lambda} = 0$. Thus, if the other parameters are identical, SIIR, SIIRS and SIIS models can have substantially different evolutionary outcomes of latency in the first infection stage, especially if transmission decays faster than progression as a function of latency.

If the progression decay exponent is greater than the transmission decay exponent, i.e. $c_2 > b_2$, then the change in evolutionary outcome due to immunity is more moderate (figure 4*c*). Indeed, suppose that with no loss of immunity there is a unique interior ESS. As the duration of immunity decreases, then the interior ESS shifts to a smaller value of latency, until eventually it reaches zero latency. Thus, for these values of the progression and transmission decay exponents, changes in immunity modulate the evolutionary dynamics of latency through incremental changes in the value of the ESS, which may eventually lead to the replacement of an interior ESS at positive latency by one at zero latency.

## 3.4. More complex evolutionary behaviour

In the model without superinfection and under the specific formulations of equations (3.1) and (3.2) for the transmission and progression rates of the first stage as a function of latency, there is either a unique interior evolutionarily singular strategy, or the unique ESS is either at zero or infinite latency. Analogous to that scenario, we have proved that with superinfection there can be a unique interior singular strategy under certain conditions, with the caveat that this singular strategy could be convergence stable and not evolutionarily stable. Thus, the existence of a unique branching point is a major contrast to the model without superinfection. To add to this difference, superinfection also gives rise to the possibility of more than one interior evolutionarily singular strategy. Indeed, our analyses that establish uniqueness do not contain all possible parameter combinations, and numerically we find alternative evolutionary behaviours. (Due to numerical constraints, in situations where uniqueness is not established, singular strategies at large $\lambda$ and those at infinity cannot be distinguished.)

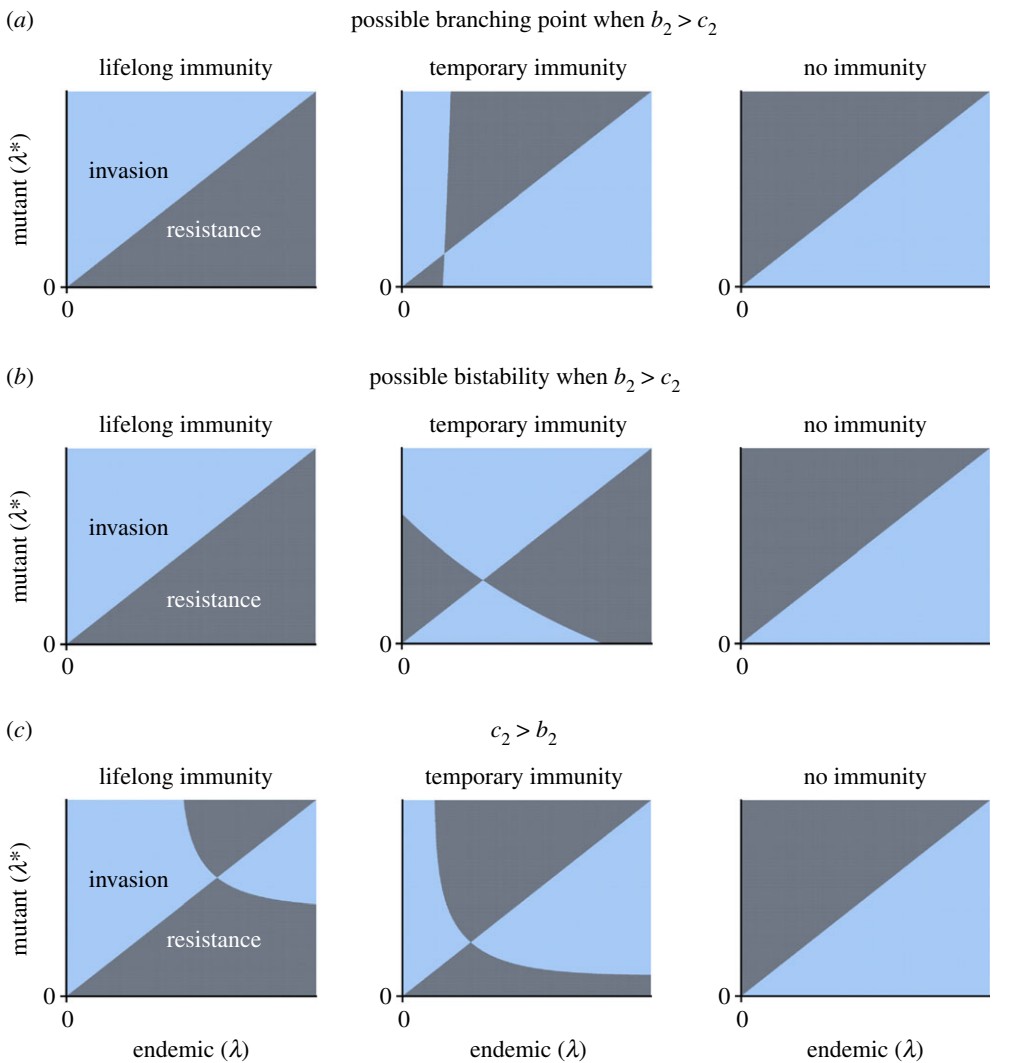

**Figure 4.** Schematics of evolutionary dynamics, mediated by host immunity. Here, the light blue regions depict successful invasion of the resident phenotype by the mutant, whereas the dark regions represent the mutant dying out and the resident remaining. Furthermore, $\sigma = 1$, i.e. individuals in the first infection stage can be infected by another pathogen at the same rate as susceptible individuals. For each panel, the leftmost, central and rightmost plots depict pairwise invasibility plots (PIPs) for the case where immunity is lifelong ($\mu = 0$), an intermediate length of immunity ($0 < \mu < \infty$), and no immunity due to infection (large $\mu$) for (a) $b_2 > c_2$ giving rise to a branching point, (b) $b_2 > c_2$ giving rise to bistability, and (c) $c_2 > b_2$. In each panel, the other parameters are fixed and chosen so that different behaviours emerge for different values of $\mu$ [for (a): $\alpha_{1,\infty} = 0.2$, $v_{1,\infty} = 0.02$, $b_1 = 0.04$, $c_1 = 0.05$, $\delta = 1/(365(50))$, $\alpha_2 = 0.24$, $\sigma = 1$, $c_2 = 0.6$, $b_2 = 0.62$, $v_2 = (\alpha_2/2) - \delta$; for (b): $\alpha_{1,\infty} = 0.2$, $v_{1,\infty} = 0.05$, $b_1 = 0.1$, $c_1 = 0.05$, $\delta = 1/(365(50))$, $v_2 = 0.1$, $\alpha_2 = 0.3$, $\sigma = 1$, $c_2 = 0.6$, $b_2 = 0.75$; for (c): $\alpha_{1,\infty} = 0.2$, $v_{1,\infty} = 0.05$, $b_1 = 0.1$, $c_1 = 0.05$, $\delta = 1/(365(50))$, $v_2 = 0.1$, $\alpha_2 = 0.3$, $\sigma = 1$, $c_2 = 0.75$, $b_2 = 0.6$]. Note that the axes values are fixed across plots for each panel, which is of particular importance for comparisons between the leftmost and central plots of (c). In (a, right), (b, right) and (c, right), we have verified that the unique ESS is at zero latency. In (a, left) and (b, left), the analysis only guarantees that there is a local ESS at infinite latency, but it seems to be unique.

Figure 5 illustrates the emergence of two interior evolutionarily singular strategies. Figure 5a has a unique attractor at infinite latency. With a large enough mutational change of latency to $\lambda^*$, however, it can be seen from the PIP that a non-monotonic approach to the ESS at infinite latency is possible, i.e. mutants with a smaller latency than the endemic strain—and far enough away from the resident phenotype—can invade. For this setting, the regions of mutual invasibility are illustrated in figure 5d. Indeed, a strategy with large latency can invade a resident with zero latency, and a resident at high latency can be invaded by a mutant with zero latency, even if the unique attractor is at maximal latency. Without superinfection, the PIP is symmetric, and so such mutual invasibility is not possible.

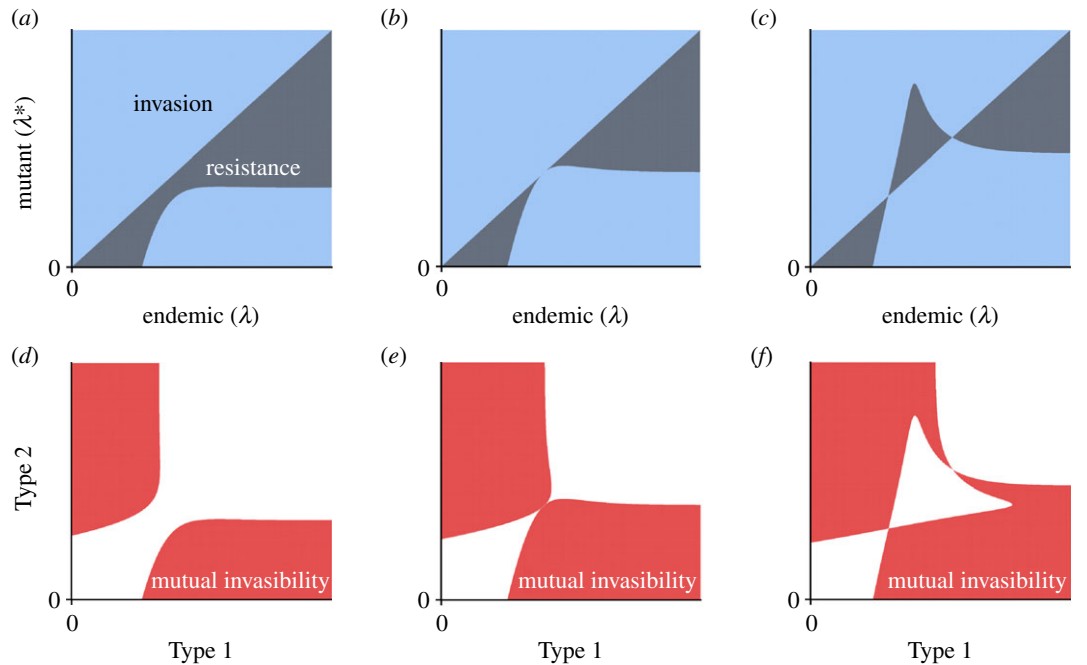

**Figure 5.** Schematics of pairwise invasibility plots (PIPs) showing the emergence of more complex phenomena with superinfection. Here, $b_2 > c_2$, and the rate of loss of immunity is chosen to highlight the contrasting evolutionary behaviours: leftmost column $\mu = 1/102 = 0.0098$, centre column $\mu = 0.0099361$, rightmost column $\mu = 1/99 = 0.0101$. (a) No interior singular strategy and so a unique ESS at infinite latency. (b) A unique interior singular strategy that is convergence semi-stable and not an ESS. (c) Two interior singular strategies, both of which are not ESSs, but one of which is convergence stable whereas the other is a repeller. (d–f) Mutual invasibility regions for the PIP illustrated in (a), (b) and (c), respectively. Here, Type 1 and Type 2 refer to the latencies of strategy 1 and strategy 2, respectively. In (a–c), the colour code is as in figure 3a and figure 4, and in (d–f), it is as in figure 3b.

On the other hand, as $\mu$ increases, it crosses a unique value for which there is a transition in evolutionary behaviour, and the critical value is presented in figure 5b. Here, there is an evolutionarily singular strategy for which the isocline does not cross the diagonal. This strategy is convergence semi-stable: for initial latency above this singular strategy, mutants with values of latency progressively further away from the singular strategy can invade, whereas for initial latencies below the singular strategy, mutants with values intermediate to the singular strategy can invade. Thus, this singular strategy is approached in evolutionary time only from below. When it is reached, then it is susceptible to invasion by mutants with larger latency. Therefore, a small mutation can lead away from this singular strategy. In this case, transmission decays faster than progression, and therefore this singular strategy is a fitness minimum and is not evolutionarily stable. Indeed, even as the resident approaches the singular strategy from below, it can be seen that types with slightly smaller latencies can successfully invade. This is partially reminiscent of a branching point, but this mutual invasibility is only true for latency values that are near and smaller than the singular strategy. Furthermore, the region of mutual invasibility is greater when compared with the previous case without the interior singular strategy (figure 5e).

This unique singular strategy then branches into two interior strategies, depicted in figure 5c. Again, since $b_2 > c_2$, these singular strategies are not ESSs and are fitness minima (electronic supplementary material, theorem 2). Visual examination reveals that one strategy (with smaller latency) is convergence stable, whereas the other (with larger latency) is not. Therefore, the convergence-stable strategy that is not an ESS is a branching point, and the other singular strategy is a repeller. For initial values of latency above the repeller, the evolutionary endpoint is at infinite latency. With such initial latencies and small mutations, the branching point is not approached in evolutionary time. This contrasts with the evolutionary dynamics with a unique branching point and highlights the important evolutionary implications of the existence of this repeller. For other initial values, the branching point is approached, but eventually nearby mutants can invade, which leads to mutual invasibility. The region of mutual invasibility takes a different shape with these two interior singular strategies (figure 5f), which illustrates the complex behaviour that results from these two fixed points. The differences

in figure 5a–c are due to the small changes in the duration of immunity following infection. Thus, host immune responses can finely modulate more complex evolutionary phenomena that only appear in the presence of superinfection.

# 4. Discussion and conclusion

We have extended the framework developed by Saad-Roy *et al.* [12] to include superinfection in an epidemiological-evolutionary model of pathogen latency. For tractability, we have assumed that superinfection leads to an immediate clearance in the host of the resident pathogen in favour of the superinfecting pathogen, and we have followed the biologically reasonable assumption that superinfection only occurs in the initial stage of an infection. Including the biological realism of superinfection, however, implies that the endemic phenotype shapes the invasion landscape of the mutant through both naive and superinfection susceptibility.

Using adaptive dynamics, we have shown that the evolutionary outcomes of latency depend upon the values of the superinfection parameter, $\sigma$, the transmission decay exponent, $b_2$, and the fully latent transmission rate times the average lifespan, $\alpha_{1,\infty}/\delta$. This is particularly the case when transmission decays faster than progression in the first stage. Indeed, we proved that in this regime, then any evolutionarily singular strategy is not an ESS, and we further showed that there can exist an ESS at zero latency, a singular strategy (possibly unique) that is a repeller and so leads to bistability at zero and maximal latency. We also showed that there can exist a unique convergence-stable singular strategy that is not an ESS, and is therefore a branching point. If progression decays faster than transmission, then any singular strategy is an ESS, and we showed that there can either be an ESS at zero latency, or an interior CSS, leading to a sub-symptomatic first infectious stage. Through numerical simulations, we have also illustrated that two interior evolutionarily singular strategies can exist with superinfection, posing a stark contrast to the case without superinfection. It is also worth noting that our 'latent' stage is distinct from a short stage after infection before a host is infectious and without symptoms, though we explore a generalization of this scenario in electronic supplementary material, *The addition of infection stages*. Furthermore, in a situation where hosts with no symptoms cannot transmit, i.e. $\alpha_{1,\infty} = 0$, then $\lambda = \infty$ is never an ESS. Intuitively, a completely presymptomatic stage during which pathogens are not transmissive does not evolutionarily emerge, nor does the branching point (irrespective of the strength of superinfection). A symptomless host lacking the ability to transmit could be due to a variety of host or pathogen constraints, and our model illustrates the evolutionary implications of such constraints.

More specifically, our underlying epidemiological model captures a continuum of immune durations. Most interestingly, we have shown that superinfection can enable the duration of host immunity to influence critically the evolutionary outcomes of latency. For example, superinfection can lead to different evolutionary possibilities if immunity is lifelong or if hosts have no immunity following recovery. By contrast, without superinfection, the SIIRS, SIIR and SIIS frameworks all give rise to identical evolutionary dynamics. This notable distinction is powerful: under superinfection, for the same shapes of trade-offs, the possible outcomes for latency evolution can be set by the duration of host immunity. Furthermore, significant qualitative and quantitative changes of evolutionary dynamics can cascade from a small change in the duration of immunity. For certain viral diseases in humans, such as influenza or coronaviruses, it is thus imperative to determine the propensity for superinfection by closely related strains, in addition to the shape of the underlying trade-off between transmission and progression. Once these are quantified, our model could be appropriately parametrized while also taking into account the duration of immunity following infection, and the predictions from our model tested. If a branching point is found to exist, our model would provide a potential mechanism to explain the coexistence of viral lineages.

Other studies on superinfection have focused on a different life-history trait (virulence) rather than latency, with different underlying epidemiological assumptions. For example, Boldin & Diekmann [35], Alizon & van Baalen [34] and Boldin *et al.* [36] assumed a single infection stage that is lifelong, and Kada & Lion [38] assumed immediate return to susceptibility following recovery. By contrast, we have investigated latency with a model that contains two infectious stages. Our model is general and includes both recovery and immunity, and we have shown the importance that host immunity can have in shaping the qualitative evolutionary dynamics of latency.

With regard to superinfection, we have so far assumed that the superinfection parameter is constant across the latency spectrum, which is the most 'minimal' model of strain replacement. In an SI model

with disease-induced death, in order for branching to emerge, the superinfection parameter as a function of both resident and mutant strategies must satisfy certain properties (given in Alizon [37]). Furthermore, with superinfection, branching can also exist in an SI model (for virulence evolution) when transmission is a function of virulence [36]. Here, we have studied another life-history trait (latency), and our underlying epidemiological model is more general than an SI model (i.e. a model in which hosts do not recover and hence also can never develop immunity following recovery). However, superinfection could itself be a function of latency, and this would incorporate further realism into our framework and would probably lead to more complicated evolutionary dynamics. In electronic supplementary material, *The superinfection parameter as a function of latency*, we show that relaxing the assumption of a constant superinfection parameter can lead to alternative evolutionary behaviour. In particular, when progression decays faster than transmission (i.e. $c_2 > b_2$), it is now possible that an interior evolutionarily singular strategy may not be an ESS.

Overall, we find that including this simplified approach to superinfection for hosts early in the course of infection (i.e. in the first infection stage, while pathogen loads are building up), can lead to numerous qualitative differences in evolutionary dynamics. In particular, a larger superinfection parameter decreases the parameter space for which there is an ESS at maximal latency (whether local or unique). However, a larger superinfection parameter also implies that a unique branching point can exist. Furthermore, it appears (numerically) that a coalition of zero and maximal latency can be evolutionarily stable. While this coalition outcome does not represent a local ESS at maximal latency, it nevertheless suggests that it may be possible for a fully latent pathogen to emerge evolutionarily even when there is high superinfection. In addition, the inclusion of superinfection in our modelling framework can also lead to multiple interior evolutionarily singular strategies.

Thus, from a public-health perspective, superinfection can have dramatic and counterintuitive evolutionary effects on latency. In order to gain pathogen-specific insight into the evolution of latency, it is therefore necessary to characterize the relative propensity of superinfection relative to susceptible infection. Then, initiatives such as the isolation of infectious individuals can be regarded with an evolutionary lens. Indeed, knowledge of the evolutionary outcomes could clarify if it is sufficient to isolate all infectious individuals together (and hence possibly enable superinfection), or if care should be taken to properly isolate individual cases (to reduce the probability of superinfection). The economic costs of these alternative measures can differ substantially, and more costly initiatives would probably only be enacted if there existed clear benefits. Thus, these questions are of significant public health interest, and they have implications for the current COVID-19 pandemic.

Under the assumption of lifelong immunity, the qualitative evolutionary dynamics of latency in the first infection stage without superinfection does not change if an additional infection stage is included in the multi-stage compartmental model [12]. Assuming that infection confers lifelong or no immunity, i.e. $\mu = 0$ or $\mu \to \infty$ (which guarantee a stable endemic equilibrium if $\mathcal{R}_0 > 1$ in either case [51,52]), analogous observations hold with superinfection. Indeed, the evolutionary dynamics of latency are qualitatively identical for an additional stage after the second infection stage, which can superinfect but cannot be superinfected, or an initial stage prior to the first stage, which cannot superinfect nor be superinfected (electronic supplementary material, *The addition of infection stages*). For example, numerous diseases exhibit a short period after infection before transmission starts.

Based on our findings on branching in our simple two-stage model with superinfection, we conjecture that even more complex evolutionary behaviour could arise if we consider more stages with different superinfection parameters. Under this framework, there may be a scenario that leads to serial branching. Such a result would be particularly appealing given that certain pathogens exhibit a range of latency in the first stage, and would indicate that a general theory could explain the emergence of all these divergent life-history strategies. Lastly, in the case without superinfection and with a similar model where $v_2$ denotes disease-induced death instead of recovery, the evolutionary dynamics of latency were found to be equivalent [12], and similar conclusions would hold here also.

In a different context, our result on branching could also be applied to pathogen speciation. Our work illustrates that superinfection mediates scenarios where, for identical parameters, pathogens with two different strategies can successfully invade each other. Furthermore, we presented a numerical example with a coexistence equilibrium and a coalition of two strategies that is evolutionarily stable with respect to mutations. If additional branching was observed in a more complicated model as we conjecture above, then this could further our understanding of the coexistence of pathogens with different strategies but similar underlying biology.

While our model assumes the simplification that there is no within-host coexistence of pathogen strains (as occurs for HIV [53]), our results may nevertheless (more speculatively) also tie to the

evolution of latency in the human immunodeficiency virus (HIV). For HIV, transmission during the latent stage is important, as has been underlined in numerous studies [5,54,55]. Here, we find a mechanism that could explain the emergence of viruses that exhibit different latencies. Indeed, if superinfection in the latent stage of HIV can occur, then the probability of superinfection is enhanced since this stage can last many years. Further, the likelihood of superinfection is also especially increased since these individuals could still engage in risky behaviour, which could heighten their exposure to a virus with another phenotype. Extensions of our evolutionary-epidemiological model to consider co-infections could elucidate the connection between HIV co-infections and the coexistence of strains with different latencies.

## 4.1. Further directions

To model superinfection, we have made two key assumptions. First, we have assumed that the superinfecting pathogen immediately replaces the one being superinfected. It would be valuable to investigate the evolutionary dynamics of latency if longer co-infection can occur. Second, motivated by underlying biological processes, we have assumed that superinfection only occurs in the initial stage of infection. However, it is possible that superinfection in the symptomatic stage could occur with a different propensity compared to that in the first infection stage, and this would also be a logical direction to explore. Investigating the addition of further infection stages with different propensities of superinfection would also be an immediate extension. Additionally, while we have explored a situation in which the superinfection parameter decays as a function of latency, it would be worthwhile to more generally characterize the evolutionary behaviour in these cases.

In our model, we have focused on pathogen properties and evolutionary dynamics, and ignored host characteristics, such as heterogeneity and their (longer) evolutionary timescales. However, the inclusion of superinfection leads to the modulation of latency evolutionary outcomes by host immunity, which suggests that specific host characteristics could affect latency evolution. One potential approach would be to perform similar latency evolutionary analyses with superinfection within the epidemiological model framework presented by Morris *et al.* [56] (see also [57]) with a secondary susceptible class for those individuals that were previously infected but whose immunity has waned. More broad natural future directions would be to investigate host–pathogen coevolutionary dynamics between pathogen latency and host immunity, or to examine the effect of individual heterogeneity in host immunity on the evolutionary dynamics of latency. Furthermore, we have focused on pathogens that infect a single host and are directly transmitted. However, to study the emergence of a presymptomatic infectious stage for other diseases caused by opportunist pathogens, our model would require the inclusion of other important factors, such as multiple hosts species and the surrounding environment (including other pathogens) [58].

Furthermore, while we have proved that a unique branching point can exist in our model, we have not analytically proved the long-term epidemiological and evolutionary behaviour after the system approaches a branching point. Most importantly, it would be valuable to show that mutual invasibility implies coexistence for our model. Furthermore, in these regions with mutual invasibility, it would be useful to show that there exists a unique coexistence equilibrium that is stable, which would then facilitate analytical characterizations of evolutionary dynamics after branching.

Our work could also be taken in a myriad of other more general directions. For example, Saad-Roy *et al.* [12] numerically examined the evolutionary dynamics of latency without superinfection under more complex transmission and progression trade-offs, and it is very likely that this complexity would translate across to this new framework. Other possible extensions discussed by these authors could also be applied with this more general framework. In particular, considering more infection stages with evolutionary dynamics that depend on the evolutionary outcome of latency in the first stage could be incorporated.

It would also be useful to explicitly incorporate vaccination in this epidemiological-evolutionary latency modelling framework. Indeed, 'leaky vaccines' (those that do not fully block transmission) or population vaccination strategies can have unexpected evolutionary and epidemiological outcomes [26,59], and it would be worthwhile to see how different vaccines and vaccination schemes interplay with the evolutionary dynamics of latency.

Lastly, while all of these prospective directions are at the population level, it would also be interesting to characterize the within-host dynamics of pathogen replication and competition. In the framework with superinfection, this would be particularly informative, and could inform the shapes of the trade-offs in addition to possibly determining the value of the superinfection parameter.

Data accessibility. Data are available in the electronic supplementary material.

Authors' contributions. Designed research (C.M.S.-R., B.T.G., S.A.L., L.P.); performed research (C.M.S.-R., S.A.L., N.S.W., P.v.d.D.); wrote the paper (C.M.S.-R., B.T.G., S.A.L., L.P., H.B.S., N.S.W., P.v.d.D.).

Competing interests. We declare we have no competing interests.

Funding. We acknowledge funding from the United States Centers for Disease Control and Prevention, the RAPIDD program of the Science and Technology Directorate Department of Homeland Security and the Fogarty International Center, NIH, the James S. McDonnell Foundation 21st Century Science Initiative Collaborative Award in Understanding Dynamic and Multi-scale Systems, the C3.ai Digital Transformation Institute and Microsoft Corporation (award no. AWD1006615), the Natural Sciences and Engineering Research Council (NSERC) of Canada (a Discovery Grant to P.v.d.D. and a Postgraduate-Doctoral Scholarship to C.M.S.-R.), and Gift from Google, LLC. This work was also supported in part by the National Science Foundation, through grant no. CNS-2027908, through the Center for the Physics of Biological Function (PHY-1734030) and through Expeditions in Computing grant no. CCF1917819. L.P. and H.B.S. gratefully acknowledge funding from the Wellcome Trust and Royal Society (grant no. 202562/Z/16/Z).

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
